# Sarcopenia as a Prognostic Factor for the Outcomes of Surgical Treatment of Colorectal Carcinoma

**DOI:** 10.3390/healthcare13070726

**Published:** 2025-03-25

**Authors:** Šimon Leščák, Martina Košíková, Sylvia Jenčová

**Affiliations:** 1Faculty of Medicine, Pavol Jozef Šafárik University in Košice, 040 11 Košice, Slovakia; lescaksimon5@gmail.com; 2Faculty of Management and Business, University of Presov, 080 01 Prešov, Slovakia; sylvia.jencova@unipo.sk

**Keywords:** sarcopenia, colorectal carcinoma, prognostic factor, surgical treatment, bibliometric analysis

## Abstract

**Background/Objectives:** Sarcopenia, defined as the progressive loss of muscle mass and function, is increasingly associated with worse outcomes in the surgical treatment of colorectal cancer (CRC). This paper focuses on analyzing the impact of sarcopenia as a prognostic factor on postoperative outcomes in CRC patients. The aim of the study is to identify the main factors influencing postoperative results. This will be accomplished via bibliometric analysis and highlighting the significance of muscle condition in the preoperative assessment of patients. **Methods:** The methodological approach involves analyzing bibliographic data from relevant scientific databases focused on sarcopenia and oncological surgery. The study employs a quantitative analysis of citations and collaborations among authors and institutions. The focus will be on research publications from 2013, when we first identified references to sarcopenia in the examined context. **Results:** The results show that sarcopenia significantly increases the risk of postoperative morbidity and mortality in CRC patients. Network analysis and keyword mapping reveal dominant research topics such as muscle condition, mortality, and postoperative complications. Meanwhile, we identify the need for standardized diagnostic methods for sarcopenia and their integration into clinical practice to improve predictive models and clinical approaches. **Conclusions:** These findings underscore the importance of interdisciplinary collaboration, preoperative assessment of muscle condition, and the implementation of standardized approaches to improve clinical outcomes for patients.

## 1. Introduction

Colorectal cancer (CRC) is among the most frequently diagnosed malignancies worldwide, and surgical resection remains a fundamental pillar of its treatment [1]. The success of surgical treatment and overall patient survival are significantly influenced by a wide range of prognostic factors, including age, comorbidities, nutritional status, and body composition [2]. Among these is sarcopenia, which is characterized by a progressive loss of muscle mass, strength, and function. Sarcopenia is increasingly recognized as a clinically relevant predictor of adverse surgical outcomes, particularly in older and oncology patients [3,4].

This syndrome arises from the multifactorial interplay of systemic inflammation and tumor metabolism. It is also adversely affected by oncological treatments such as chemotherapy and radiotherapy [3]. In colorectal cancer surgery, sarcopenia has been linked to higher rates of postoperative complications. These complications include prolonged hospitalization and reduced survival, underscoring its importance as a prognostic factor.

The main mechanisms underlying the development of sarcopenia are as follows [3,5,6,7,8,9]:Metabolic Changes Induced by Tumors and Imbalance in Muscle Protein Synthesis and Degradation—Aging as well as tumor diseases lead to a decline in anabolic signaling pathways. One such pathway is the insulin-like growth factor 1 (IGF-1) pathway, which is essential for muscle growth. Simultaneously, there is an increase in catabolic factors, such as myostatin and inflammatory cytokines (e.g., TNF-α, IL-6). These factors promote proteolysis and muscle degradation. This process can be exacerbated by chronic inflammation, resulting in increased production of pro-inflammatory cytokines. It can also accelerate protein breakdown, ultimately leading to muscle mass loss.Systemic Inflammation—Tumor diseases often induce a chronic inflammatory state, leading to increased production of pro-inflammatory cytokines such as interleukin-6 (IL-6) and tumor necrosis factor-alpha (TNF-α). These molecules contribute to the activation of the proteasome system and the degradation of muscle proteins. Inflammatory cytokines enhance the activity of proteolytic pathways. This results in protein degradation and muscle mass loss. This inflammatory process can be exacerbated by conditions such as obesity and chronic diseases, further accelerating muscle loss. Inflammation also increases the activity of the proteasome system, which regulates the breakdown of muscle proteins.Oxidative Stress and Mitochondrial Dysfunction—Oxidative stress, caused by an imbalance between the production of reactive oxygen species (ROS) and antioxidant defense mechanisms, damages muscle cells. This impairs their function and contributes to sarcopenia. Increased oxidative stress can activate pathways, leading to muscle fiber atrophy and apoptosis. Moreover, mitochondrial dysfunction, which affects the energy metabolism of muscles, can worsen oxidative stress and inflammation in muscle tissues. This further promotes muscle degradation.Inadequate Nutrient Intake—Anorexia and malnutrition, common in patients with colorectal carcinoma, significantly reduce the availability of amino acids necessary for muscle tissue regeneration. Insufficient protein intake disrupts the balance between protein synthesis and proteolysis, leading to muscle atrophy. Malnutrition can also impair the body’s ability to synthesize new muscle proteins and maintain existing muscle mass.Physiological Aging—In older patients, sarcopenia is exacerbated by a natural decline in muscle mass due to hormonal changes, such as decreased testosterone and growth hormone levels, as well as reduced physical activity. Aging increases the accumulation of fat tissue within muscles (myosteatosis). It also affects the energy metabolism of muscle cells. Both of these aging-related factors promote muscle degradation processes. In elderly individuals and those with chronic diseases, satellite cell function may decline, reducing their ability to regenerate muscle tissue. This reduced muscle regeneration contributes to the muscle mass loss characteristic of sarcopenia.

In oncology patients, sarcopenia often develops due to a combination of the aforementioned factors, along with the side effects of treatment. These side effects include fatigue, anorexia, the toxic effects of chemotherapy, and reduced mobility during treatment. These factors lead to decreased physical performance, increased postoperative complications, prolonged hospitalization, and reduced overall survival [4]. Understanding these mechanisms provides insight into potential therapeutic strategies for preventing or mitigating sarcopenia. This is particularly true in patients undergoing surgical treatment for colorectal carcinoma. Addressing factors such as nutrition, inflammation, and physical activity may improve the outcomes in this patient population.

In recent years, sarcopenia has gained attention not only as an indicator of physical frailty but also as a significant predictor of postoperative complications, long-term survival, and patient quality of life. The development of imaging technologies and the implementation of standardized diagnostic criteria (such as skeletal muscle analysis using CT scans) have enabled a more in-depth analysis of its impact on treatment outcomes. However, many questions remain regarding the interactions between sarcopenia, other clinical factors, and patient management strategies. Colorectal cancer remains a major global health concern, with significant implications for surgical and oncologic outcomes. The increasing recognition of sarcopenia as a prognostic factor in CRC has led to a growing body of research exploring its role in risk stratification and patient management. Given the well-documented impact of sarcopenia on postoperative complications and survival, a systematic understanding of its research landscape is essential for improving clinical decision-making and future therapeutic strategies.

Accurate diagnosis of sarcopenia plays a crucial role in clinical decision-making, particularly in the preoperative assessment and management of colorectal cancer (CRC) patients. Early identification of muscle mass depletion allows for timely intervention strategies, such as prehabilitation programs, nutritional support, and targeted exercise regimens. These interventions can enhance surgical outcomes and reduce postoperative complications. Integrating standardized sarcopenia assessment into routine oncological evaluations could contribute to more personalized treatment planning, helping clinicians optimize perioperative risk stratification and improve patient prognosis.

This study aims to analyze the impact of sarcopenia as a prognostic factor on postoperative outcomes in CRC patients. Through bibliometric analysis, the study seeks to examine the evolution and current state of research, identify key factors affecting postoperative outcomes, and highlight the significance of muscle condition in preoperative patient assessment. The study also aims to identify major research trends, dominant themes, research gaps, and collaborative links between authors to contribute to a better understanding of the dynamics and potential of sarcopenia as a prognostic factor in CRC surgical treatment outcomes.

## 2. Materials and Methods

Bibliometric analysis is a quantitative approach to examining scientific publications. This analysis enables a systematic literature review and evaluation using computational tools. This method is used to map relationships between authors, analyze citation rates, assess keyword occurrences, and explore thematic areas. Its goal is to identify trends and dominant themes. This in turn should allow the identification of connections within academic research in the examined field.

In the bibliometric analysis, we included only publications in which sarcopenia was directly analyzed as a prognostic factor in the context of CRC surgical treatment. Studies were identified in which sarcopenia was the primary variable or explicitly discussed as a key factor influencing postoperative outcomes. In cases where sarcopenia appeared only as a secondary research aspect (e.g., in broader analyses of comorbidities or nutritional status), these publications underwent additional evaluation for relevance. Only those studies in which its impact on surgical outcomes was analyzed using quantitative methods or statistically confirmed were included in the final dataset.

Bibliometric analysis provides significant advantages, such as offering a systematic overview of the literature, allowing for the rapid identification of current trends, key concepts, and leading experts in the research field. It also supports the visualization of relationships between topics and authors, contributing to a better understanding of collaboration dynamics and scientific networks. Another key benefit is the identification of research gaps, helping to uncover underexplored areas with the potential for further investigation. These insights not only serve as a foundation for developing new studies but also act as a strategic tool for research project planning [10].

In the context of colorectal cancer surgical treatment and its prognostic factors, bibliometric analysis serves as a valuable tool for assessing the current state of research. It allows for the identification of key research themes and understanding trends in the use of indicators such as sarcopenia for predicting treatment outcomes and long-term patient prognosis.

### Research Questions

Based on the study objectives, the following research questions were formulated:RQ1. How has research on sarcopenia as a prognostic factor in surgical treatment outcomes for colorectal carcinoma evolved?RQ2. What are the main themes and research directions in publications focusing on the impact of sarcopenia on CRC surgical treatment outcomes?RQ3. What connections exist between authors, their countries of origin, and their publications in the study of sarcopenia in oncologic surgery?RQ4. Which keywords and concepts dominate research on sarcopenia and its impact on CRC surgical treatment outcomes?

The Web of Science database was used to identify and extract relevant scientific publications. Web of Science is a recognized academic and research source covering a wide range of scientific journals and articles. The bibliometric analysis included publications—scientific articles exported from Web of Science—retrieved using a Boolean query in the “Topic” search field to capture various synonyms and related terms for the study topic: (“sarcopenia” OR “muscle wasting” OR “skeletal muscle depletion” OR “muscle loss” OR “low muscle mass”) AND (“colorectal cancer” OR “colorectal carcinoma” OR “colon cancer” OR “rectal cancer” OR “CRC” OR “colorectal neoplasm”) AND (“surgery” OR “surgical treatment” OR “surgical resection” OR “colectomy” OR “rectal resection”) AND (“prognosis” OR “prognostic factor” OR “outcome” OR “surgical outcome” OR “postoperative complications” OR “survival rate” OR “treatment outcome”).

For processing and visualizing the bibliometric analysis results, VOSviewer software (version 1.6.20) was used. This tool enables the creation of network maps based on citations, author collaborations, or keyword frequency. Using VOSviewer, it was possible to identify thematic clusters, key research directions, and connections between individual studies.

To deepen the understanding of the topics examined, a direct literature review was conducted after the bibliometric analysis. This process focused on articles labeled “Highly Cited Papers” in Web of Science, which reflect the most influential contributions and perspectives in the field. Detailed analysis of these studies provided deeper insights into the main themes, methodologies, and conclusions that serve as a basis for future research on sarcopenia as a prognostic factor in the outcomes of colorectal cancer surgical treatment. The identified findings are expected to contribute to the improvement of clinical management strategies and the prognostic evaluation of patients.

As part of the direct review of the most frequently cited articles, the WordSift platform was also utilized. This tool was used for analyzing and visualizing key terms and topics appearing in the texts of the most cited publications in the field of sarcopenia as a prognostic factor in CRC surgical treatment outcomes.

WordSift provides tools for extracting significant words and phrases from a text. This enables the quick identification of dominant themes within the research field. When a text is inserted into the tool, a visualization is generated. This displays words based on their frequency of occurrence, helping to reveal the most frequently discussed concepts and trends.

These visual representations facilitated a better understanding of the structure of the highly cited articles and allowed for the identification of key terms and methodological approaches considered most relevant in this field of research. This method enriched the review process with a visual analytical component. This simplified the interpretation of textual data and supported a higher-quality assessment of research trends in sarcopenia-related studies.

## 3. Results

Research on the impact of sarcopenia on the outcomes of surgical treatment for CRC represents a crucial and growing area of clinical and scientific interest. As the number of studies highlighting the prognostic significance of sarcopenia in postoperative outcomes increases, it becomes necessary to systematically analyze this development to better understand its clinical implications. The aim of this study is to systematically analyze the evolution of knowledge regarding sarcopenia as a prognostic factor in patients undergoing CRC surgery. The goal is to examine the current state of scientific knowledge in this area, identify key factors influencing postoperative outcomes, discover dominant research themes and existing gaps, and explore collaboration patterns among authors and institutions. This analysis contributes to a deeper understanding of the importance of muscle condition in the preoperative management of patients and provides a foundation for further clinical research and the optimization of surgical care for CRC patients.

For greater clarity in presenting the findings, the results were divided into three main parts. The introductory section of this chapter is devoted to a quantitative analysis of scientific publications, addressing the predefined research questions. The analysis of publications from the Web of Science database responds to the first two research questions: (1) How has research on sarcopenia as a prognostic factor for surgical outcomes of colorectal cancer evolved? (or more generally, how has research on prognostic factors for surgical outcomes of CRC evolved?) and (2) What are the main research directions in publications focused on the impact of sarcopenia on CRC surgical outcomes? For comparison, two data sets were analyzed—one specifically targeting sarcopenia (the search query is provided in the methodology) and another encompassing more general prognostic factors for CRC surgery without explicitly mentioning sarcopenia.

The second part then presents a graphical visualization of the interrelationships among authors, countries of origin for the publications, and dominant keywords, thereby answering the third and fourth research questions. The final and third part of the results focuses on a detailed literature analysis of selected highly cited publications.

### 3.1. General Information from Bibliometric Analysis

Based on the identification and extraction of relevant scientific publications that met the criteria outlined in the methodology section, a total of 238 scientific publications were retrieved. These publications contained the specified search terms in their titles, abstracts, or author keywords (as of 10 January 2025).

An overview of the distribution of the number of publications by year and document type is presented in Table 1 below.

In response to the first research question (RQ1), there has been a growing interest in the topic of sarcopenia as a prognostic factor for surgical outcomes in CRC, with the earliest identified publications dating back to 2013. This relatively short timeframe suggests that sarcopenia has only begun to appear in the context of CRC surgical treatment in recent years. The most common document types were original research articles and review papers, while the number of letters to the editor and corrections was minimal. A noticeable increase in publications is observed from 2020 onward. These peaked in 2024 (42 publications), with original research articles predominating during that peak (39 out of 42 in 2024). The year 2025 was not included in the analysis because the literature search was conducted in early January 2025. This rise in the number of publications since 2020 may be linked to heightened awareness of the role of muscle condition in patient prognosis, as well as advancements in imaging technologies used to diagnose sarcopenia.

A graphical representation of the publication trends in this area is shown in Figure 1. The trend line reflects increasing scientific interest in examining the impact of sarcopenia on prognosis and surgical outcomes in colorectal cancer. Over the years, one can observe shifts in the nature of publications. Theoretical and review-oriented articles predominated in the early years. Clinical and empirical studies focusing on prognostic factors and survival analysis have become more prevalent in recent years. From a temporal standpoint, there is a clear transition from fundamental observations to large-scale meta-analyses and the application of advanced technologies. This illustrates the progressive understanding of sarcopenia as a significant factor affecting CRC treatment outcomes.

Analysis of the chronological development of publications reveals several trends:
From 2013 to 2017, studies began to appear in the literature focusing on the relationship between sarcopenia and postoperative complications in CRC patients. These works primarily addressed the basic concept of sarcopenia, including its definition and diagnostic approaches through CT imaging. Also, many demonstrated the negative impact of muscle mass depletion on surgical outcomes [11,12,13,14].Among the key studies is the work by the authors of [15], which examined the link between sarcopenia and postoperative complications after cytoreductive surgery with hyperthermic intraperitoneal chemotherapy; their findings showed that sarcopenia increases the risk of severe complications. Similarly, ref. [16] reported a negative effect of sarcopenia on survival following curative resection for CRC. Several studies also focused on morphometric analysis of muscle mass [17,18,19]. Ref. [20] demonstrated that muscle density is a strong predictor of postoperative complications. And ref. [21], in a systematic review, confirmed that a smaller muscle mass—measured via CT—is associated with a higher risk of complications and elevated mortality.Research on the psoas muscle as an indicator of sarcopenia has also emerged. For instance, ref. [22] investigated changes in the psoas muscle and their relationship to postoperative complications. And ref. [23] explored the influence of the psoas muscle on the occurrence of postoperative infections. Ref. [24] further confirmed that low muscle mass measured on CT increases the risk of postoperative complications and infections, emphasizing the importance of psoas muscle assessment when evaluating surgical risk.Other studies [25,26] showed that laparoscopic resection can be performed on sarcopenic patients without significantly increasing complications and that the ERAS protocol can reduce the negative impact of sarcopenia on short-term outcomes and functional recovery.From 2018 to 2020, research on sarcopenia in the CRC context shifted from the basic concept and diagnostics to a deeper examination of its specific aspects and clinical significance. Publications began to focus more on particular facets of sarcopenia, such as its relationship to myosteatosis (excess fat infiltration in muscles) [27,28,29,30,31,32] and visceral obesity [33,34,35]. These studies investigated how the combination of sarcopenia, myosteatosis, and visceral obesity affects postoperative outcomes and patient survival.During this period, there was also a growing number of studies evaluating various methods of measuring muscle mass and their accuracy [29,36,37,38,39,40,41]. In addition to CT scans—considered the gold standard—other methods, such as bioelectrical impedance analysis (BIA), were compared, and different indices like the psoas index (PI), the L3 skeletal muscle index (L3SMI), and the psoas cross-sectional area were assessed.A push toward integrating sarcopenia measurements into clinical practice can also be observed [42,43,44,45]. These studies investigate how sarcopenia metrics can aid in predicting postoperative complications, hospital length of stay, and overall survival, which could facilitate better treatment planning and improve patient outcomes.Besides overall survival, research also addresses the impact of sarcopenia on laparoscopic surgery outcomes [46,47,48], on neoadjuvant and adjuvant therapies [49,50,51,52], and on patients with metastases as well as older patient populations [53,54,55,56]. Findings from individual studies suggest that sarcopenia can also negatively affect these dimensions of treatment.Some research [57,58] explores how prehabilitation—a combination of nutritional and exercise interventions before surgery—can improve muscle mass and thus enhance postoperative outcomes. This area shows promising potential for improving treatment results.Current Period (2021–2024/2025)—Recent research on sarcopenia in the context of CRC has become increasingly in-depth and expansive, emphasizing the clinical application of findings and the identification of new prognostic markers. Studies from this period build on previous evidence that sarcopenia serves as a predictor of postoperative complications and survival, while introducing fresh insights into its dynamics, interactions with other factors, and potential therapeutic approaches.One key development involves the examination of sarcopenia dynamic changes in muscle mass before and after surgery—and its effect on survival, with persistent postoperative sarcopenia negatively impacting overall survival [59,60]. Researchers have also explored the role of sarcopenia in exacerbating inflammatory processes before and after surgery. Sarcopenia was found to amplify inflammatory responses, further worsening prognosis and complications [61]. However, the inflammatory role in sarcopenia is not entirely clear, necessitating a comprehensive approach that includes nutrition, exercise, and anti-inflammatory therapy [62].In the context of neoadjuvant therapy, studies have focused on sarcopenia’s impact on the response and toxicity of chemoradiotherapy (NACRT), finding that its presence can diminish treatment efficacy and heighten side effects [63,64,65,66]. A multidisciplinary team (MDT) approach has emerged as beneficial in reducing postoperative complications and shortening hospital stays in CRC patients, potentially yielding positive outcomes for sarcopenic patients as well [67]. Research has also assessed the influence of sarcopenia on robotic surgery outcomes, revealing higher complication rates and lower survival even with this surgical modality [68].Additional investigations have highlighted the connection between sarcopenia and socioeconomic and environmental factors [69]. Findings suggest that socioeconomic deprivation may increase the prevalence of sarcopenia and myosteatosis, thereby impairing CRC prognosis [70]. Diagnostic studies have turned to developing novel methods and indices for assessing sarcopenia, including artificial intelligence (AI), to analyze CT images and predict complication risks [71,72,73,74,75,76,77]. Emphasis has been placed on psoas muscle density and the visceral-to-subcutaneous fat (V/P) ratio as predictors of postoperative outcomes and survival [78,79,80,81,82,83,84].Several studies have examined the preoperative nutritional factors (sarcopenia, osteosarcopenia, malnutrition, obesity, dietary inflammatory potential) affecting postoperative CRC outcomes. These studies show that sarcopenia is an independent risk factor for complications and prolonged hospitalization [85,86,87,88,89,90]. Researchers have also evaluated the synergistic effects of sarcopenia with factors such as anemia [91], aerobic fitness [92], malnutrition [93,94,95,96], inflammatory markers [86,97,98], sarcopenic obesity [99,100,101,102,103,104], age [105,106,107,108], and frailty [109,110] on survival and morbidity.In the realm of prehabilitation and rehabilitation, studies have investigated the benefits of multimodal prehabilitation [111,112,113], nutritional status, and the efficacy of nutritional supplements [114,115,116], as well as neuromuscular electrical stimulation (NMES) [117] in reducing postoperative sarcopenia. Besides complications and survival, researchers have also examined quality of life in post-operative CRC patients [118].Findings from this period confirm that sarcopenia is an independent risk factor for postoperative complications, reduced survival, and poorer responses to treatment in CRC patients. Importantly, not only muscle quantity but also muscle quality (density) is crucial. However, minimally invasive surgery may mitigate some of the negative effects of sarcopenia and myosteatosis [119,120]. Across these studies, the importance of preoperative screening for sarcopenia [68,121,122,123,124] and the need for a personalized treatment approach [125,126]—accounting for muscle mass, overall health, and social factors—are strongly emphasized. Nevertheless, standardizing diagnostic methods and definitions of sarcopenia remains essential for a more effective comparison of study outcomes [127,128,129,130].

A descriptive time series analysis was conducted to examine the temporal development of publications on sarcopenia as a prognostic factor in the surgical treatment of colorectal cancer. The marked increase in publications since 2020 indicates a progressive expansion of research in this area, with a peak recorded in 2024. This timeline was evaluated through trend analysis, focusing not only on the rise in the number of publications but also on the shift from theoretical to empirical clinical studies.

For comparison, Table 2 and Figure 2 depict the publication trends relating to colorectal cancer and its surgical treatment without including sarcopenia. Compared to Figure 1, this visualization underscores the significantly higher volume of publications in this domain, with research dating back to 1975. The broader thematic scope resulted in a wider variety of document types, including conference abstracts, editorials, and book chapters. The rapid increase in publications post-2010 suggests an expansion of prognostic factor analysis, reflecting advancements in surgical techniques, risk stratification, and patient outcomes.

Over time, the nature of published outputs has evolved from general studies on CRC prognostic factors to specialized research on muscle mass and its impact on postoperative outcomes. This reflects a growing interest in this specific factor within oncological surgery. Figure 2 illustrates the point at which sarcopenia began appearing in scientific articles as a factor studied in prognostic outcomes for colorectal cancer surgery. In earlier periods, sarcopenia was not addressed in the scientific literature. As the years progressed, it became an increasingly analyzed prognostic factor, underscoring its rising importance in clinical research.

Based on the quantitative analysis of publications and the identification of key works, we now move on to a more detailed examination of the thematic focus within the Web of Science categories. This provides a more comprehensive perspective on the issue under study. Figure 3 visualizes the distribution of research areas and addresses research question RQ2.

In the field of sarcopenia research as a prognostic factor in the surgical treatment of colorectal cancer, the largest concentration of publications is in the Surgery category, with 112 publications. This high figure reflects the close connection between sarcopenia research and surgical approaches in colorectal cancer treatment. Another significant category is Oncology, with 73 publications, confirming that sarcopenia is an important prognostic factor for oncology patients, especially regarding cancers of the colon and rectum.

Other notable areas, such as Gastroenterology Hepatology (40 publications) and Nutrition Dietetics (33 publications), underscore the importance of nutrition and gastroenterological perspectives in understanding the relationship between sarcopenia and the overall condition of patients with colorectal cancer. Categories like Medicine General Internal (24 publications) and Geriatrics Gerontology (14 publications) indicate that sarcopenia is frequently studied in the context of older patients, aligning with its prevalence in geriatric populations undergoing surgery for colorectal cancer. Additionally, fields such as Medicine Research Experimental (6 publications) and Rehabilitation (5 publications) show smaller but ongoing research efforts focusing on experimental and rehabilitative strategies for colorectal cancer patients affected by sarcopenia.

Compared to broader prognostic factors for surgical outcomes in colorectal cancer (see Table 3), there is a significant difference in the number of publications; however, Surgery and Oncology continue to dominate in that broader category as well. These findings suggest that surgical and oncological elements are the most frequently investigated in the context of predicting treatment outcomes—consistent with the trends observed in sarcopenia research.

An interesting finding is that fields such as Medicine General Internal (1296 publications) and Radiology Nuclear Medicine Medical Imaging (539 publications) appear in large numbers. This highlights the importance of diagnostic techniques and internal factors in assessing prognosis.

By comparing the Web of Science research areas identified through both search queries, studies on sarcopenia as a prognostic factor are strongly represented in surgical and oncological disciplines. This aligns with research focused on forecasting surgical outcomes. In contrast, prognostic factors for colorectal cancer are explored across a broader range of areas, including gastroenterology, internal medicine, and diagnostic fields. All of these areas are relevant to the comprehensive assessment of patients. In addition to sarcopenia, numerous other factors have been identified that could, alongside sarcopenia, negatively impact colorectal cancer surgical outcomes or increase the risk of sarcopenia—subsequently affecting postoperative results. These include [102] age [105,107,108], sex [106], low body mass index (BMI), low preoperative albumin levels, smoking, tumor size and stage [131], body composition (encompassing sarcopenia, myosteatosis, and visceral obesity), patient functional status (e.g., ASA score, frailty), low subcutaneous fat, poor aerobic capacity [92], diabetes, and other comorbidities. These factors underscore the complexity of preoperative evaluations and the need for an individualized approach to managing patients with colorectal cancer.

For a more comprehensive view of global research on this issue, we next focus on the geographical distribution of publications. Within the study of sarcopenia as a prognostic factor in colorectal cancer surgical outcomes, we identified 38 countries whose authors addressed this topic. However, most of these countries have a low number of publications, indicating limited global interest in this specific area. The countries with the largest number of scientific contributions include China (48) and Japan (42), which, together with the Netherlands (27), account for one-third of the publications. Figure 4 shows the countries that contributed the most to scientific research in this field. Slovakia is represented by just one publication. Darker shades on the map indicate countries with a higher volume of publications on sarcopenia as a prognostic factor in CRC surgical outcomes, with China (48) and Japan (42) leading the research output. The lighter shades represent countries with fewer contributions, emphasizing the regional disparities in scientific engagement with this topic. Zero occurrence is displayed in gray. It is important to note that the geographic distribution of publications does not necessarily reflect biological differences in the prevalence of sarcopenia across populations. Rather, it may be influenced by research priorities, the availability of diagnostic technologies, and the level of clinical interest in specific regions. Some studies suggest that there may be variations in muscle composition and sarcopenia prevalence across different ethnic groups due to genetic, nutritional, and lifestyle factors. However, this study does not directly analyze racial or ethnic differences in sarcopenia, but focuses on the bibliometric analysis of research outputs in this field. Therefore, the conclusions drawn from Figure 4 should be interpreted as representing the intensity of scientific activity rather than inherent racial predispositions to sarcopenia.

Figure 5, which presents the results for the search query examining prognostic factors for the surgical outcomes of colorectal cancer without a narrower focus on sarcopenia, reveals significantly higher publication activity. The largest number of publications comes from China (3206), the USA (3053), and Japan (2621)—countries traditionally characterized by high research productivity. Slovakia has 14 publications in this domain, a markedly higher number compared to those presented in the more narrowly focused first table. The wider global participation in this category highlights the extensive exploration of diverse prognostic variables beyond sarcopenia. However, it should be noted that these publications span a longer period, dating back to 1975.

### 3.2. Network Analysis of Co-Authorship and Keywords

As mentioned in the introduction to the results section, after applying the constraints specified in the methodology, 238 articles focusing on sarcopenia as a prognostic factor in the surgical treatment of CRC by country were included in the bibliometric analysis, along with 20,055 publications targeting prognostic factors for surgical treatment of CRC more broadly. Given the limitations on the number of articles extractable into the VOSviewer tool and considering the theme of the present paper, this part of the results focuses solely on the 238 publications centered on sarcopenia.

Within the bibliographic analysis using VOSviewer, co-authorship was analyzed first from the perspective of the countries engaged in this issue, addressing Research Question 3 (RQ3). From the publications examined, 39 countries were identified in total (the United Kingdom was represented as England, Wales, and Scotland; Turkey was listed under both Turkey and Turkiye). However, several countries were not interlinked (Denmark, Finland, Greece, Indonesia, Israel, Lebanon, Mexico, New Zealand, Poland, Portugal, Romania, Singapore, South Africa, Spain, Taiwan, and Turkey). To enhance interpretability, only the largest connected set of countries was selected.

Figure 6 shows a bibliometric map that classifies six color-differentiated clusters of 20 collaborating countries. The size of each node corresponds to the publication volume, while the thickness of the connecting lines indicates the strength of collaborative ties. The country with the highest number of publications is China (48 publications, 3 links, total link strength 3), followed by Japan (42, 1, 1) and the Netherlands (27, 5, 7). These numbers indicate that while China and Japan lead in publication output, they engage fewer in international collaborations compared to their European counterparts. Conversely, the Netherlands exhibits the highest citation count (1607), suggesting its influential role in this research domain. This is followed by Japan (1090) and China (882). The classification of countries into clusters is as follows:Red cluster (4 countries)—England, Japan, Scotland, WalesGreen cluster (4 countries)—Brazil, Italy, Jordan, SwitzerlandDark blue cluster (3 countries)—Australia, Iran, NetherlandsYellow cluster (3 countries)—Canada, Ireland, SloveniaPurple cluster (3 countries)—France, South Korea, USALight blue cluster (3 countries)—Austria, Germany, China

Methodologically, the classification was based on link strength and the minimum number of publications. The colors represent different clusters, and the size of the nodes corresponds to the publication activity of individual countries. The bibliometric map was generated using VOSviewer, with restrictions applied regarding the minimum threshold of collaboration between countries.

As part of the bibliometric analysis using VOSviewer software, co-authorship among researchers in this field was also examined. A total of 1677 authors were identified. However, for better interpretability, the analysis included only those authors with a minimum of three publications. This threshold was met by 62 authors, but not all of them were connected to the other researchers. The largest set of interconnected authors consists of 16 individuals, as shown in the bibliometric map in Figure 7.

The co-authorship map reveals three distinct clusters, each represented by a different color:The red cluster includes authors with strong mutual connections, focusing on similar research topics. This cluster is characterized by a high level of internal collaboration, suggesting a specialized research focus.The green cluster represents another group of collaborating authors, with multiple connections among its members. This cluster primarily focuses on specific aspects of sarcopenia research and surgical oncology.The blue cluster consists of a smaller number of authors with weaker connections to other clusters, yet forming a cohesive research network. This cluster is linked to the red cluster through a few key authors, indicating an interdisciplinary connection among researchers.

The size of the nodes on the map corresponds to the number of publications by each author, while the thickness of the links between them indicates the intensity of their collaboration. Among the most prominent authors in the analyzed field are predominantly Chinese and Japanese researchers, who lead scientific production in this domain.

The overall collaboration network suggests that research on sarcopenia as a prognostic factor in the surgical treatment of colorectal cancer is largely centralized within certain research groups and has a regional character. Asian countries are playing a dominant role.

Beyond individual co-authorship networks, several large-scale international collaborations have contributed significantly to advancing sarcopenia research. Notable among these is the collaboration between researchers in Japan and China. Leading institutions, such as the University of Tokyo and Fudan University, have been instrumental in conducting high-impact studies on sarcopenia’s role in colorectal cancer prognosis.

Furthermore, interdisciplinary collaborations between oncologists, nutritionists, and rehabilitation specialists have emerged. This is exemplified by multi-center studies investigating prehabilitation protocols for sarcopenic cancer patients. These studies, often coordinated by European and North American research groups, have explored the integration of nutritional supplementation, resistance training, and multimodal rehabilitation to mitigate the adverse effects of sarcopenia on surgical outcomes.

The presence of distinct research clusters suggests that, while significant progress has been made in understanding sarcopenia in colorectal cancer, further efforts to enhance global research integration could yield more standardized diagnostic approaches and optimized treatment strategies.

Within the keyword analysis of research on sarcopenia as a prognostic factor for the outcome of surgical treatment of colorectal cancer (RQ4), a total of 845 keywords were identified (442 author keywords and 498 KeyWords Plus). To enhance interpretability, only those keywords appearing at least 10 times in the publications were included in the analysis. The resulting bibliometric map in Figure 8 presents a network of keywords grouped into four thematically distinct clusters, each represented by a different color. Each cluster reflects a specific thematic area of research. The classification of the 57 keywords is as follows:Red cluster (20 keywords): This cluster, focused on clinical and methodological factors related to surgical treatment, includes the following terms: classification, colorectal surgery, consensus, frailty, impact, index, infection, inflammation, morbidity, mortality, muscle mass, outcomes, postoperative complications, predictor, prehabilitation, recovery, resection, risk factors, sarcopenia, surgical complications.Green cluster (14 keywords): It reflects physical parameters and imaging methods associated with sarcopenia and oncological diseases, featuring terms such as body composition, body mass index, cancer, colon, colorectal-cancer, complications, computed-tomography, malnutrition, obesity, rectal cancer, risk, sarcopenic obesity, skeletal-muscle, surgery.Blue cluster (13 keywords): This group centers on prognostic factors and treatment outcomes of CRC, with terms such as chemotherapy, clinical implications, colon cancer, colorectal cancer, curative resection, diagnosis, mass, prevalence, prognosis, skeletal–muscle mass, solid tumors, survival, toxicity.Yellow cluster (10 keywords): Oriented toward specific patient conditions and risk factors associated with sarcopenia, this cluster includes terms like body composition, cachexia, depletion, elderly patients, meta-analysis, myosteatosis, prognostic factor, short-term outcomes, tomography, visceral obesity.

Larger nodes on the map represent keywords with a higher frequency of occurrence, while thicker connections indicate a stronger co-occurrence between terms in the analyzed publications. The visualization enables an overview of dominant research themes and their interconnections, illustrating how various aspects of sarcopenia are related within the scientific literature.

Among the most frequently used keywords (considering all keywords) were sarcopenia (occurrences: 197, total link strength: 2374), resection (80, 962), colorectal cancer (74, 875), surgery (73, 896), complications (67, 771), outcomes (66, 780), impact (63, 735), and survival (60, 747).

The results of this study demonstrate that research on sarcopenia as a prognostic factor for outcomes of surgical treatment of colorectal cancer is broadly diverse, encompassing various aspects of clinical research, predictive factors, and surgical outcomes. The network analysis identified key thematic areas related to sarcopenia, including muscle mass, postoperative complications, mortality, and oncological outcomes. An examination of international collaboration among countries highlighted the dominant roles of China, Japan, and the Netherlands in terms of publication output and citation impact. Keyword mapping revealed four main thematic clusters focusing on risk classification, body composition, chemotherapy, and prognostic factors. The interconnections among these clusters suggest strong interdisciplinary collaboration and a continued interest in enhancing predictive models for CRC surgical treatment.

### 3.3. Overview of the Most Cited Scientific Publications on Sarcopenia as a Prognostic Factor in Surgical Treatment of Colorectal Cancer

The number of citations of scientific publications in the field of sarcopenia as a prognostic factor for colorectal cancer surgical outcomes reflects not only the degree to which the scientific community accepts these findings, but also their significance in the context of subsequent research. A high citation count may indicate that a given article has made a key contribution to the advancement of knowledge and serves as a major reference point for future studies. Conversely, publications with fewer citations may suggest limited visibility or a less groundbreaking nature. However, visibility and citation numbers can be influenced by various factors, such as the length of time since the article was published, the scope and quality of the databases in which it is indexed, and the prevailing thematic interest at a given time.

Within this bibliometric analysis, a review was conducted of the four most cited scientific publications from the Web of Science database focusing on sarcopenia as a prognostic factor in the surgical treatment of colorectal cancer. The goal of this analysis is to identify the main scientific perspectives on which past research has concentrated and to highlight key thematic areas that may be the subjects of future investigations.

The most frequently cited scientific publication (383 citations) at the time of analysis, registered in the Web of Science database, is titled “Functional Compromise Reflected by Sarcopenia, Frailty, and Nutritional Depletion Predicts Adverse Postoperative Outcome After Colorectal Cancer Surgery”. In this study, ref. [14] examines the impact of functional impairment—represented by sarcopenia, frailty, and nutritional deficits—on postoperative outcomes among patients undergoing surgical treatment for colorectal cancer. The findings indicate that a combination of a high Groningen Frailty Indicator score, a high Short Nutritional Assessment Questionnaire score, and the presence of sarcopenia is a significant predictor of postoperative sepsis.

Included with this brief overview of articles are visualizations created using the WordSift platform, which display the keywords and topics appearing in the most cited publications on sarcopenia as a prognostic factor in colorectal cancer surgery. These images show words and terms according to their frequency in the text, enabling quick identification of dominant concepts and trends in a given topic and publication. In the case of the first analyzed article, after inputting the title, abstract, and keywords into the tool, the visualization (Figure 9) reveals the most frequently discussed terms—such as “sarcopenia”, “frailty”, “functional”, and “nutritional”—which dominate in the context of examining the influence of functional impairment on postoperative complications.

This publication makes a significant contribution to the issue of sarcopenia as a prognostic factor in surgical oncology. The study provides a detailed analysis of the clinical implications of functional impairment. And it also offers a comprehensive model of predictive indicators that could be useful for further clinical research and the optimization of preoperative care.

Ranked second by citation count (278), the 2015 article titled “Sarcopenia is a Negative Prognostic Factor After Curative Resection of Colorectal Cancer” [16] examines the impact of skeletal muscle depletion (sarcopenia) on the prognosis of colorectal cancer patients after curative resection. The results showed that patients with sarcopenia had significantly shorter recurrence-free survival (RFS) and overall survival (OS) compared to those without this condition.

The keyword cloud for the analyzed publication, displayed in Figure 10, identifies the most frequent terms, such as “sarcopenia”, “patient”, “RFS”, “skeletal”, “muscle”, and “survival”. These terms dominate in the context of investigating the effects of sarcopenia on surgical treatment outcomes. The study underscores the importance of early identification of sarcopenia as an independent prognostic factor using skeletal muscle quantification based on preoperative CT scans.

Miyamoto’s work analyzes the macro-level impact of sarcopenia on patient survival following surgery. Our presented study focuses on a bibliometric analysis of the scientific literature examining trends and research orientations in the field of sarcopenia as a prognostic factor in colorectal oncology. These approaches complement each other, providing a comprehensive view of the significance of sarcopenia in clinical practice.

The third publication in terms of citation count (206 citations in the Web of Science database) is titled “Sarcopenia and cachexia in the era of obesity: clinical and nutritional impact”. In this study, ref. [13] investigates the variability of body composition in modern populations using imaging techniques and explores its relationship with clinical outcomes. Particular attention is paid to sarcopenic obesity—conditions characterized by low muscle mass and excessive adipose tissue—that can affect the prognosis of patients with various diseases, including cancer.

The authors found that sarcopenia and sarcopenic obesity can lead to increased chemotherapy toxicity, shorter time to tumor progression, worse postoperative outcomes, and overall reduced survival rates. They emphasized the importance of nutritional assessment and the need for targeted interventions to optimize body composition in this heterogeneous patient group.

The keyword cloud for the analyzed publication, shown in Figure 11, reveals the most frequent terms, such as “obesity”, “cancer”, “sarcopenia”, “body”, “muscle”, and “nutritional”. These concepts reflect the study’s focus on the connection between body composition and clinical outcomes. Ref. [13] highlights the need for a differentiated approach to nutritional requirements—especially in terms of protein and energy intake—given that the states of sarcopenia and sarcopenic obesity create distinct metabolic demands.

A study by the authors of [13] is relevant to our research because it addresses sarcopenia and sarcopenic obesity as risk factors for poorer clinical outcomes, which may significantly affect the prognosis of patients with colorectal cancer (CRC). Although ref. [13] primarily focuses on the impact of body composition on cancer treatment outcomes, their research provides important insights that expand our understanding of how sarcopenia influences patients with CRC. This is especially true in the context of predicting survival and surgical outcomes. This approach considers the complexity of sarcopenia as a predictor in clinical decision-making, which supports our analysis in the field of surgical treatment of colorectal cancer.

The fourth analyzed scientific publication, titled “Sarcopenia predicts worse postoperative outcomes and decreased survival rates in patients with colorectal cancer: a systematic review and meta-analysis”, had 135 citations in the Web of Science database at the time of our study. This systematic review, which analyzed 44 studies, shows that sarcopenia is a strong predictor of worse postoperative outcomes and reduced survival in colorectal cancer (CRC) patients. In its meta-analysis, ref. [132] found that patients with sarcopenia are at higher risk of postoperative complications, such as infections, cardiopulmonary issues, and prolonged hospital stays. Furthermore, sarcopenia is associated with decreased overall survival, disease-free survival, and cancer-specific survival. This publication is relevant to our research on sarcopenia as a prognostic factor in the surgical treatment of CRC. The study underscores the importance of preoperative assessment and the need to consider sarcopenia when planning treatment for CRC patients.

The keyword cloud identified using the WordSift tool (Figure 12) includes words like “postoperative”, “survival”, “patient”, “cancer”, “sarcopenia”, “outcome”, and “sarcopenic”, which repeatedly appear in the analyzed literature and are crucial for understanding prognostic factors in this context.

The publications discussed in this bibliometric analysis offer a comprehensive perspective on various aspects of sarcopenia as a prognostic factor for the outcomes of surgical treatment in colorectal cancer. These studies highlight the importance of key clinical variables—such as functional impairment, nutritional status, muscle mass, and patients’ metabolic characteristics—that influence postoperative results. The presented article enhances our understanding of how these factors can predict risks and optimize surgical care management. It also underscores the challenges associated with integrating them into clinical practice.

## 4. Discussion

Research on the impact of sarcopenia on the outcomes of surgical treatment for CRC represents a key and growing area of clinical and scientific interest. With an increasing number of studies highlighting the prognostic significance of sarcopenia in postoperative outcomes, it is essential to systematically analyze this development to gain a better understanding of its clinical implications.

The results of this bibliometric analysis confirm that sarcopenia is an independent prognostic factor that significantly affects postoperative outcomes in CRC patients. Several high-impact studies have consistently demonstrated that sarcopenia is associated with increased postoperative complications, higher morbidity, and reduced overall survival in CRC patients. For example, a systematic review by [132] confirmed that sarcopenia is a strong predictor of worse postoperative outcomes, including prolonged hospital stays and increased rates of infectious complications. Additionally, research by [16] found that CRC patients with sarcopenia had significantly shorter recurrence-free survival and overall survival compared to non-sarcopenic patients. These findings align with our bibliometric trends, emphasizing the need for early identification and intervention strategies for sarcopenia in surgical oncology. Findings indicate that low muscle mass is associated with an increased risk of complications, prolonged hospitalization, and reduced overall survival, which aligns with previous publications, e.g., [60,62]. A crucial aspect of our approach was the use of studies explicitly testing and quantifying the relationship between sarcopenia and surgical outcomes in CRC. As multiple studies have pointed out, although some research has analyzed the impact of sarcopenia as part of broader studies focusing on comorbidities or nutritional status, they often lack robust quantitative data to draw definitive conclusions about its independent prognostic significance [132]. This approach allowed us to identify key trends and challenges in this field while emphasizing the need for standardized diagnostic methods.

The identified thematic clusters and key research directions suggest that risk classification, body composition, chemotherapy, and prognostic factors dominate the field. The analysis also uncovered significant collaborative links among authors and countries, highlighting strong interdisciplinary collaboration and a persistent interest in enhancing predictive models for CRC surgery. Moreover, the review of scientific literature indicates that sarcopenia is not only a critical prognostic factor for colorectal cancer but also for other areas of surgical oncology.

A chronological analysis of publications suggests that after 2020, there has been a significant increase in research focused on sarcopenia in CRC patients. This trend can be attributed to several factors, including advancements in imaging technologies (e.g., enhanced quantification of muscle mass using CT) and growing awareness of the importance of frailty and muscle condition in the surgical management of oncology patients [59,119]. Additionally, this surge in research may be linked to a broader recognition of multimodal prehabilitation approaches, which emphasize the role of nutritional and physical interventions before surgery. The inclusion of sarcopenia assessments in clinical guidelines and the growing integration of AI in imaging diagnostics have further contributed to increased attention in this field. The increased attention to sarcopenia reflects a shift from fundamental research defining this phenomenon to more practically oriented studies examining its clinical implications and potential for therapeutic intervention optimization [68].

A well-structured MDT approach is essential for optimizing the management of sarcopenia in CRC patients. Effective MDT collaboration involves oncologists, surgeons, nutritionists, and physiotherapists working in a coordinated manner to assess and mitigate sarcopenia-related risks. However, barriers to this integration include inconsistent screening practices across departments, lack of standardized protocols for prehabilitation, and logistical constraints, such as limited access to physiotherapy and nutritional counseling in some healthcare settings. Streamlining MDT communication, implementing shared digital patient records, and integrating sarcopenia assessment into preoperative guidelines may help overcome these barriers and improve patient outcomes [133,134].

Given the well-documented impact of sarcopenia on surgical outcomes, preoperative screening for muscle depletion should be integrated into routine oncological assessments. The most widely used methods for assessing sarcopenia include CT-based skeletal muscle index (SMI), which quantifies muscle mass at the L3 vertebral level [118], bioelectrical impedance analysis (BIA), which estimates muscle composition through electrical conductivity [115], and dual-energy X-ray absorptiometry (DXA), which provides a detailed assessment of lean body mass [113]. CT-based measurements are considered the gold standard due to their precision and clinical relevance in CRC patients. Beyond its prognostic significance, early identification of patients with reduced muscle mass allows for the implementation of targeted perioperative care. Standardized diagnostic criteria, such as measuring the skeletal muscle index (SMI) via imaging techniques, represent a crucial tool for surgical risk stratification. Such integration into preoperative evaluation not only helps identify high-risk patients but also facilitates interventions such as nutritional supplementation, resistance training, and anti-inflammatory therapies, which can enhance muscle functional capacity and reduce postoperative complications [8,73]. Similar recommendations can be found in review studies emphasizing the need for prehabilitation programs aimed at optimizing patients before surgery [57,111].

Recent research highlights the importance of multimodal interventions in mitigating sarcopenia-related risks. Nutritional strategies such as oral nutritional supplements (ONS) containing high-protein formulas, branched-chain amino acids, and omega-3 fatty acids have shown efficacy in improving muscle mass and reducing postoperative complications [116]. Additionally, resistance training, either alone or in combination with prehabilitation programs, has demonstrated benefits in enhancing functional recovery [113]. NMES has emerged as a potential adjunctive therapy, particularly in patients with severe sarcopenia who are unable to engage in conventional physical exercise [117]. Future clinical protocols should integrate these targeted interventions to optimize surgical outcomes and improve long-term prognosis in CRC patients. To provide a clearer overview of these therapeutic approaches, we summarize them in the following table (Table 4).

Another important aspect of our analysis is the role of systemic inflammation in sarcopenia. Increased inflammatory activity can exacerbate muscle loss and negatively impact treatment response, including chemotherapy [61,63,64]. Moreover, the connection between myosteatosis (fat infiltration of muscles) and sarcopenia further complicates the clinical picture. This highlights the need for multimodal strategies to mitigate these adverse effects [14,16].

Despite the growing evidence that links sarcopenia with adverse surgical outcomes, substantial research gaps persist. This is particularly true regarding the mechanisms underlying these relationships and the potential interventions to enhance muscle mass and function in affected patients. Future studies should focus on clarifying these mechanisms and examining targeted rehabilitation strategies to improve recovery and survival in CRC patients with sarcopenia.

Regarding future research directions, it is evident that we need large-scale prospective clinical studies to evaluate the effectiveness of intervention strategies for patients with sarcopenia. In addition to prehabilitation, emphasis should also be placed on investigating pharmacological and anti-inflammatory therapies that could modify the negative impact of muscle depletion. The integration of AI and machine learning in imaging data analysis presents a promising pathway toward automated muscle quantification and personalized risk stratification, potentially leading to evidence-based protocols incorporated into surgical oncology [119].

Finally, it is essential to emphasize that an individualized approach to CRC patients with sarcopenia should consider not only age, gender, and nutritional status but also comorbidities and other clinical factors [123,125]. Standardizing diagnostic criteria and developing a universal classification system are key steps toward improving the comparability of results across studies and optimizing perioperative care. In this context, the practical application of these findings in clinical practice is also crucial. Early identification of sarcopenia allows for the implementation of personalized prehabilitation programs, which directly contribute to improved patient outcomes and long-term prognosis [8,73].

### Limitations and Future Research Directions

While this study provides valuable insights into publication trends and key research topics, certain limitations must be acknowledged. First, bibliometric analyses are inherently limited by database selection and indexing biases. Although Web of Science is a highly reputable source, relevant publications indexed in other databases (e.g., Scopus, PubMed) may have been omitted. Second, this study focuses on publication metrics rather than clinical trial data, limiting its applicability to direct patient outcomes.

Although a strong association between sarcopenia and poor postoperative outcomes in CRC patients has been demonstrated in numerous studies, establishing direct causality remains challenging. Potential confounding factors—including preoperative nutritional status, tumor burden, inflammatory markers, and variations in treatment strategies—must be accounted for in study designs. Recent research has employed advanced statistical techniques, such as propensity score matching and multivariate regression analyses, to adjust for these confounders. However, further prospective cohort studies and randomized controlled trials (RCTs) are needed to more definitively establish causality and refine risk stratification models [14,135].

Future research should prioritize large-scale prospective clinical trials evaluating the effectiveness of targeted interventions for CRC patients with sarcopenia. While systematic reviews and meta-analyses can provide valuable insights into the clinical impact of sarcopenia, the primary objective of this study was to conduct a bibliometric analysis. This approach allows for an in-depth evaluation of publication trends, research gaps, and key contributing authors in the field. However, as part of our ongoing research, we plan to conduct a systematic review and meta-analysis on this topic in a future manuscript or monograph, which will allow for a more detailed synthesis of clinical outcomes and treatment effectiveness. Additionally, AI-driven imaging data analysis has the potential for automated muscle quantification and risk stratification. This offers a promising pathway toward personalized therapeutic approaches. Future research should also explore the integration of AI in sarcopenia diagnostics. This could be accomplished through deep learning-based analysis of CT and MRI scans. This could enhance the accuracy of muscle mass assessment. Additionally, identifying novel biomarkers that predict sarcopenia-related complications in CRC patients remains an underexplored area that warrants further investigation.

## 5. Conclusions

This paper centers on analyzing the impact of sarcopenia as a prognostic factor on postoperative outcomes in patients with CRC. Through a bibliometric analysis, the study aimed to explore the development and current state of research, identify the main factors influencing postoperative outcomes in CRC patients, and emphasize the importance of muscle condition in preoperative patient evaluation. By employing VOSviewer software and the WordSift platform, it was possible to visualize relationships among authors, the frequency of keyword usage, and the dominant topics. This combination of quantitative and qualitative approaches provided deep insight into current trends and made it possible to identify research gaps.

The results indicate a continuous increase in interest in this topic, as evidenced by a rise in publications and a diversification of research approaches. Research on sarcopenia as a prognostic factor in the surgical treatment of colorectal cancer is broadly extensive, encompassing various aspects of clinical research, predictive factors, and surgical outcomes.

A key takeaway from this study is that sarcopenia significantly affects postoperative outcomes, including increased complication rates, prolonged hospitalization, and reduced survival. Given this impact, preoperative screening for muscle depletion should be a standard component of oncological assessments. To enhance early detection and management, healthcare providers should receive targeted training on sarcopenia screening tools and risk stratification. Continuing medical education (CME) programs, interactive workshops, and the integration of artificial intelligence-assisted imaging tools could improve provider proficiency in recognizing sarcopenia. Additionally, incorporating sarcopenia assessment into standardized CRC care guidelines—such as mandatory preoperative CT-based muscle mass evaluation or functional mobility tests—could facilitate systematic risk stratification and early intervention [3]. Implementing targeted interventions such as nutritional supplementation, resistance training, and multimodal prehabilitation programs, can help mitigate the negative effects of sarcopenia and improve patient outcomes.

Furthermore, the increasing role of AI in medical imaging analysis offers new opportunities for automated muscle mass quantification and personalized risk stratification. Future research should prioritize large-scale clinical trials evaluating the effectiveness of interventions aimed at improving muscle function in CRC patients. Future studies should also prioritize multicenter prospective trials to evaluate standardized screening tools for sarcopenia in CRC surgical patients. Additionally, randomized controlled trials (RCTs) assessing the efficacy of multimodal prehabilitation interventions, including resistance training, nutritional optimization, and neuromuscular electrical stimulation, are crucial for establishing evidence-based treatment protocols.

For future research and clinical practice, standardizing diagnostic methods and implementing personalized treatment approaches are also vital. Research should concentrate on developing new therapeutic approaches, optimizing preoperative care, and utilizing MDT models to enhance survival and quality of life in CRC patients.

These findings contribute to a deeper understanding of sarcopenia’s impact on surgical outcomes and emphasize the need for a comprehensive and individualized approach to treating patients with colorectal cancer.

To optimize patient outcomes, future research should focus on implementing standardized sarcopenia screening in CRC surgical protocols, enhancing interdisciplinary collaboration through structured MDT models, and developing targeted prehabilitation strategies. Standardized diagnostic methods—such as CT-based muscle quantification—should be incorporated into routine clinical workflows. Moreover, interdisciplinary efforts between oncologists, surgeons, physiotherapists, and nutritionists should be strengthened to ensure a holistic approach to sarcopenia management. Future studies should explore the integration of AI-driven imaging for more precise risk stratification and the role of pharmacological interventions in mitigating muscle depletion. These steps are crucial for improving the prognosis and quality of life of CRC patients undergoing surgery [136,137].

## Figures and Tables

**Figure 1 healthcare-13-00726-f001:**
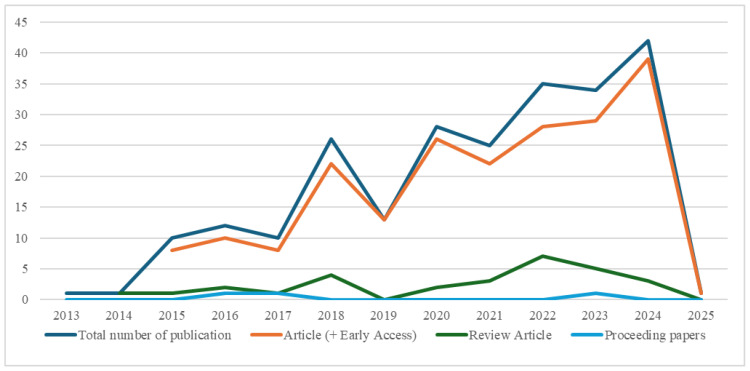
Trend of scientific publications examining sarcopenia as a prognostic factor for surgical treatment outcomes of colorectal cancer.

**Figure 2 healthcare-13-00726-f002:**
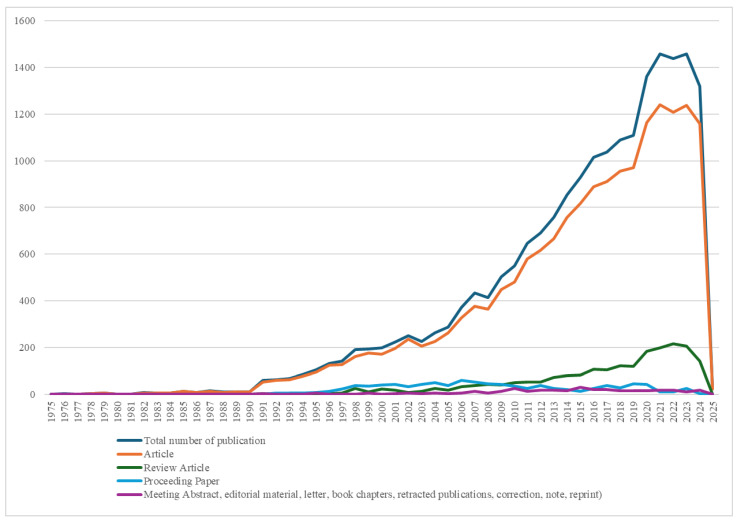
Trend of scientific publications on prognostic factors for outcomes of surgical treatment of colorectal cancer.

**Figure 3 healthcare-13-00726-f003:**
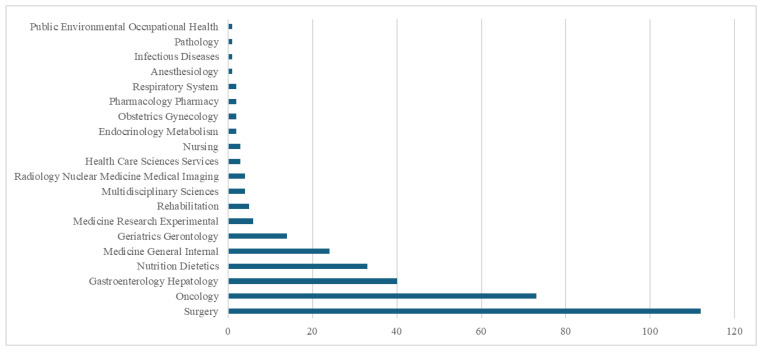
Web of Science research areas encompassing the identified scientific publications examining sarcopenia as a prognostic factor for surgical treatment outcomes of colorectal cancer.

**Figure 4 healthcare-13-00726-f004:**
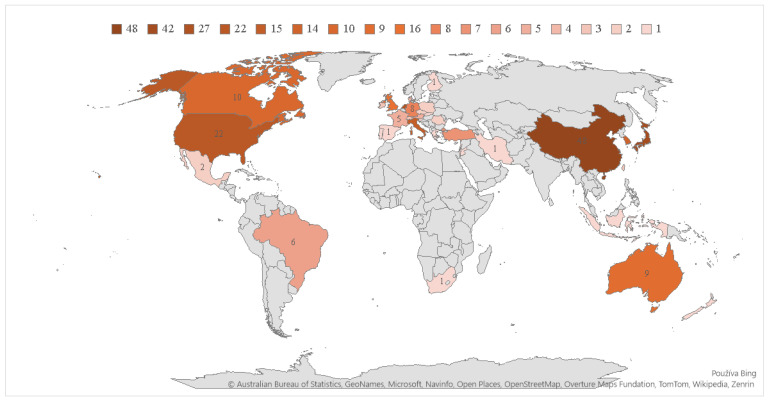
Number of publications on sarcopenia as a prognostic factor for surgical treatment outcomes of colorectal cancer by country.

**Figure 5 healthcare-13-00726-f005:**
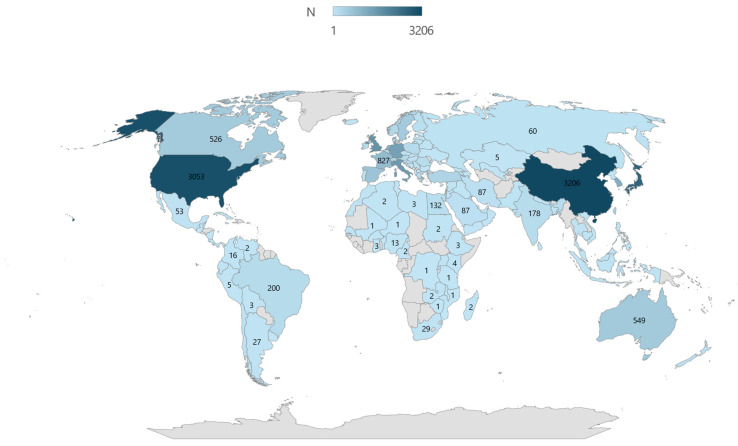
Number of publications on prognostic factors for outcomes of surgical treatment of colorectal cancer by country.

**Figure 6 healthcare-13-00726-f006:**
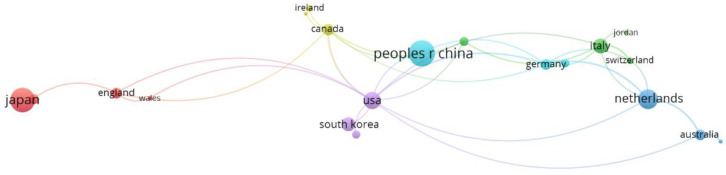
Bibliometric co-authorship map by country.

**Figure 7 healthcare-13-00726-f007:**
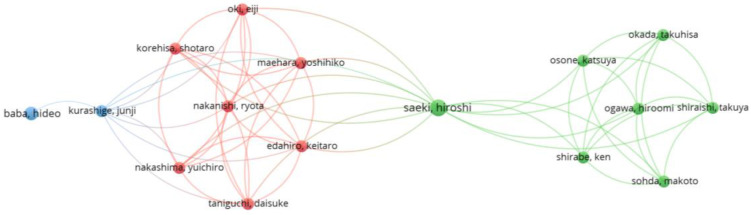
Bibliometric co-authorship map by author.

**Figure 8 healthcare-13-00726-f008:**
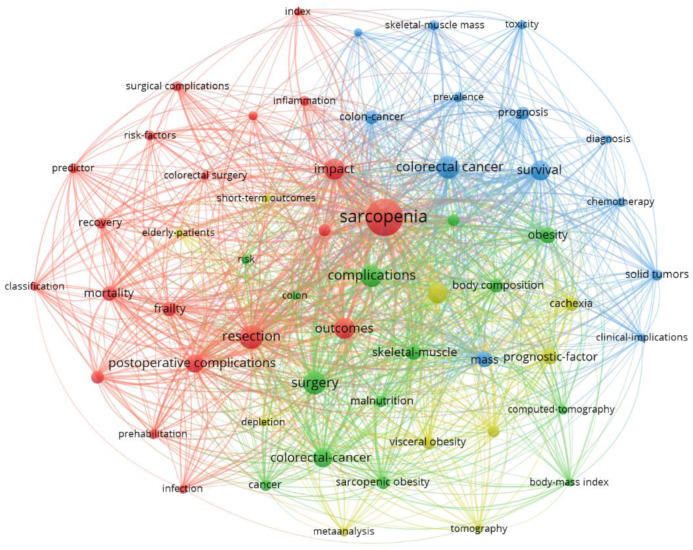
Bibliometric map of keyword occurrences.

**Figure 9 healthcare-13-00726-f009:**
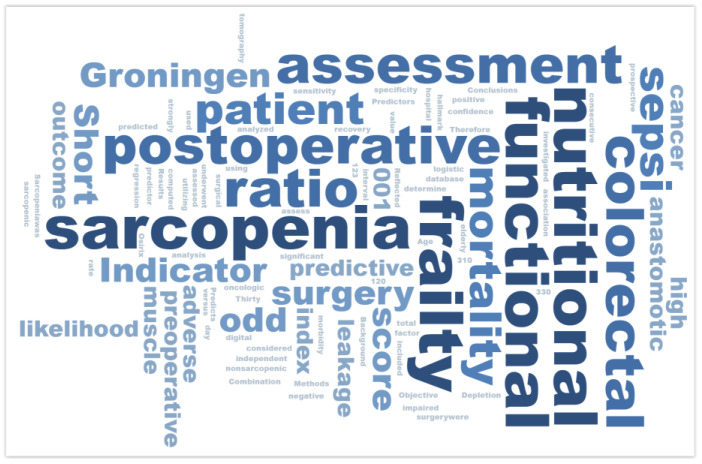
Keyword cloud in the research on sarcopenia as a prognostic factor for surgical treatment outcomes of colorectal cancer in the publication by [14].

**Figure 10 healthcare-13-00726-f010:**
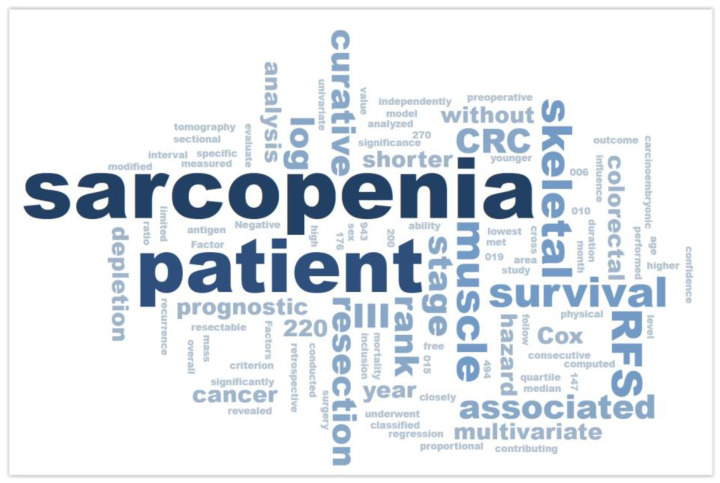
Keyword cloud for research on sarcopenia as a prognostic factor in surgical treatment outcomes of colorectal cancer in the publication by [16].

**Figure 11 healthcare-13-00726-f011:**
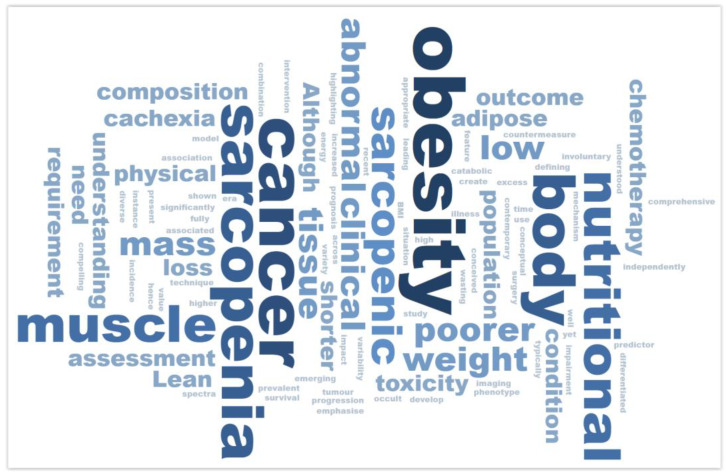
Keyword cloud for research on sarcopenia as a prognostic factor in surgical treatment outcomes of colorectal cancer from the publication by [13].

**Figure 12 healthcare-13-00726-f012:**
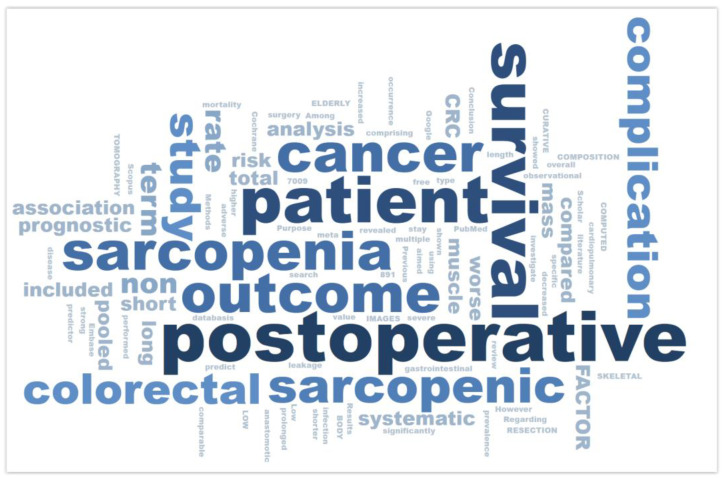
Keyword cloud for research on sarcopenia as a prognostic factor in surgical treatment outcomes of colorectal cancer from the publication by [132].

**Table 1 healthcare-13-00726-t001:** Number of scientific publications examining sarcopenia as a prognostic factor in the outcomes of surgical treatment for colorectal carcinoma.

Publication Year	Total Number of Publications	Article (Including Early Access)	Review Article	Proceeding Papers	Letter	Correction
2025	1	1	0	0	0	0
2024	42	39	3	0	0	0
2023	34	29	5	1	0	0
2022	35	28	7	0	0	0
2021	25	22	3	0	0	0
2020	28	26	2	0	0	0
2019	13	13	0	0	0	0
2018	26	22	4	0	0	0
2017	10	8	1	1	1	0
2016	12	10	2	1	0	0
2015	10	8	1	0	0	1
2014	1	0	1	0	0	0
2013	1	1	0	0	0	0
Total	238	207	29	3	1	1

**Table 2 healthcare-13-00726-t002:** Number of scientific publications examining prognostic factors for outcomes of surgical treatment of colorectal cancer.

Year	Count	Article	Review Article	Proceeding Paper	Meeting Abstract	Early Access	Editorial Material	Letter	Book Chapters	Retracted Publication	Correction Note Reprint
2025	26	24	2	0	0	2	0	0	0	0	0
2024	1319	1158	142	4	3	93	10	5	0	0	0
2023	1457	1238	207	27	4	18	4	1	0	0	3
2022	1437	1207	216	12	1	5	6	4	0	6	2
2021	1457	1240	199	10	6	3	7	3	1	2	0
2020	1362	1164	183	44	13	0	2	0	0	0	0
2019	1110	971	121	46	8	0	2	3	2	0	1
2018	1090	956	122	29	8	0	1	2	3	1	0
2017	1036	910	104	38	4	0	10	5	1	1	1
2016	1015	889	108	27	10	0	2	5	3	1	1
2015	928	816	83	13	16	0	7	3	2	2	2
2014	854	759	81	20	6	0	6	2	0	1	0
2013	759	667	73	25	10	0	4	3	1	0	0
2012	691	616	54	38	7	0	8	3	0	0	0
2011	646	580	54	27	5	0	2	4	2	0	1
2010	551	482	50	35	9	0	9	1	5	1	0
2009	503	449	40	44	3	0	4	5	2	0	0
2008	414	364	42	46	5	0	1	0	0	0	0
2007	433	378	39	54	8	0	6	0	0	0	0
2006	371	327	34	61	1	0	3	3	0	0	0
2005	288	262	19	38	3	0	0	0	1	0	0
2004	263	227	25	50	6	0	1	0	0	0	0
2003	226	206	14	44	1	0	3	0	0	0	0
2002	251	237	8	34	0	0	3	1	0	1	0
2001	223	197	18	42	1	0	3	0	0	0	0
2000	200	172	23	40	0	0	1	1	0	0	0
1999	194	177	11	35	3	0	2	0	0	0	0
1998	191	161	25	37	0	0	0	1	0	0	1
1997	143	126	7	24	1	0	1	0	0	0	0
1996	132	125	4	13	0	0	1	0	0	0	0
1995	104	94	2	9	1	0	0	4	0	0	0
1994	84	78	3	6	1	0	0	0	0	0	1
1993	67	62	2	5	0	0	0	0	0	0	2
1992	62	60	2	5	0	0	0	0	0	0	0
1991	60	53	4	1	1	0	0	0	0	0	2
1990	10	10	0	0	0	0	0	0	0	0	0
1989	10	10	0	0	0	0	0	0	0	0	0
1988	11	9	1	0	1	0	0	0	0	0	0
1987	16	14	1	0	0	0	0	0	0	0	1
1986	9	8	1	0	0	0	0	0	0	0	0
1985	14	13	0	0	0	0	0	0	0	0	1
1984	6	6	0	0	0	0	0	0	0	0	0
1983	7	7	0	0	0	0	0	0	0	0	0
1982	8	7	0	0	1	0	0	0	0	0	0
1981	2	2	0	0	0	0	0	0	0	0	0
1980	1	1	0	0	0	0	0	0	0	0	0
1979	5	5	0	0	0	0	0	0	0	0	0
1978	4	4	0	0	0	0	0	0	0	0	0
1977	1	1	0	0	0	0	0	0	0	0	0
1976	3	2	0	0	0	0	0	0	0	0	1
1975	1	1	0	0	0	0	0	0	0	0	0
∑	20,055	17,532	2124	983	147	121	109	59	23	16	20

**Table 3 healthcare-13-00726-t003:** Web of Science research areas encompassing the identified scientific publications examining prognostic factors for outcomes of surgical treatment of colorectal cancer.

WOS Category	N	WOS Category	N
Surgery	10,037	Transplantation	14
Oncology	6706	Dermatology	13
Gastroenterology Hepatology	4930	Anatomy Morphology	12
Medicine General Internal	1296	Materials Science Multidisciplinary	11
Radiology Nuclear Medicine Medical Imaging	539	Toxicology	11
Medicine Research Experimental	470	Psychiatry	10
Multidisciplinary Sciences	286	Sport Sciences	10
Pathology	263	Environmental Sciences	9
Pharmacology Pharmacy	251	Education Scientific Disciplines	8
Cell Biology	228	Economics	6
Health Care Sciences Services	161	Engineering Electrical Electronic	6
Nutrition Dietetics	156	Food Science Technology	6
Biochemistry Molecular Biology	153	Materials Science Biomaterials	6
Respiratory System	145	Physics Applied	6
Obstetrics Gynecology	120	Social Sciences Biomedical	6
Immunology	119	Anthropology	5
Public Environmental Occupational Health	112	Chemistry Analytical	5
Geriatrics Gerontology	105	Ophthalmology	5
Anesthesiology	102	Computer Science Theory Methods	4
Cardiac Cardiovascular Systems	92	Instruments Instrumentation	4
Hematology	88	Optics	4
Biotechnology Applied Microbiology	87	Tropical Medicine	4
Urology Nephrology	76	Veterinary Sciences	4
Orthopedics	72	Acoustics	3
Genetics Heredity	62	Andrology	3
Rehabilitation	55	Cell Tissue Engineering	3
Dentistry Oral Surgery Medicine	55	Chemistry Applied	3
Clinical Neurology	54	Computer Science Artificial Intelligence	3
Nursing	53	Computer Science Information Systems	3
Chemistry Multidisciplinary	51	Computer Science Interdisciplinary Applications	3
Health Policy Services	40	Imaging Science Photographic Technology	3
Medical Laboratory Technology	32	Primary Health Care	3
Endocrinology Metabolism	32	Psychology	3
Pediatrics	30	Statistics Probability	3
Critical Care Medicine	30	Automation Control Systems	2
Peripheral Vascular Disease	29	Chemistry Physical	2
Biology	29	Engineering Chemical	2
Otorhinolaryngology	27	Engineering Multidisciplinary	2
Microbiology	27	Robotics	2
Emergency Medicine	26	Audiology Speech Language Pathology	1
Developmental Biology	25	Chemistry Organic	1
Biochemical Research Methods	21	Computer Science Cybernetics	1
Integrative Complementary Medicine	20	Engineering Environmental	1
Medical Informatics	19	Management	1
Gerontology	19	Nuclear Science Technology	1
Neurosciences	18	Physics Condensed Matter	1
Mathematical Computational Biology	18	Plant Sciences	1
Engineering Biomedical	18	Polymer Science	1
Physiology	17	Psychology Multidisciplinary	1
Infectious Diseases	17	Remote Sensing	1
Biophysics	16	Social Sciences Interdisciplinary	1
Nanoscience Nanotechnology	16	Spectroscopy	1
Chemistry Medicinal	15	Water Resources	1
Reproductive Biology	14	Women’s Studies	1

**Table 4 healthcare-13-00726-t004:** Overview of therapeutic strategies for sarcopenia in colorectal cancer.

Intervention	Description	Expected Benefit	References
Oral Nutritional Supplements	High-protein supplements, BCAA, omega-3 fatty acids	Improved muscle protein synthesis, reduced inflammation	[116]
Prehabilitation	Combination of nutritional support and physical activity before surgery	Increased functional reserve, reduced postoperative complications	[58]
Resistance Training	Targeted strength exercises to maintain muscle mass	Increased muscle strength, faster recovery	[113]
NMES	Electrical stimulation of muscles in inactive patients	Preservation of muscle mass in patients with limited mobility	[117]

## Data Availability

For requests concerning the data, please contact the corresponding author.

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
