# Peer review of "Sarcopenia as a Prognostic Factor for the Outcomes of Surgical Treatment of Colorectal Carcinoma"

_healthcare, 2025, doi:10.3390/healthcare13070726_

Round 1
Reviewer 1 Report
Comments and Suggestions for Authors
Lescak et al. conducted a review on the topic of sarcopenia as a prognostic factor for CRC surgical outcomes. Strengths of the study include robust methodology and adequate number of publications included. This is an increasingly relevant topic with several clinical implications.
Would recommend to always refer to a study by Last name of first author et al rather than citation (e.g., line 614)
Would recommend a more defined breakdown of sections including separate results and discussion sections and a more concise conclusions section with bulk of that discussion moved to discussion section
- The authors did a commendable job capturing a broad range of publications on the topic. Despite this effort, one could imagine that many publications included discussions of sarcopenia and CRC surgical outcomes as secondary analyses. How did the authors address this limitation in the methods or discussion section?
Reformatting sentences to be more concise and less repetitive.
Author Response
Dear reviewer,
Please see the attachment. Thank for your valuable comments.

Reviewer 2 Report
Comments and Suggestions for Authors
Dear Authors
The manuscript explored the functional role of sarcopenia as a prognostic factor for the outcomes of surgical treatment of colorectal carcinoma. They suggested that sarcopenia’s impact on surgical outcomes.
There are number of major revision, formal and scientific aspects that should be addressed.
- According to Figure 4, is it possible that sarcopenia is race related? Therefore, it is necessary to add the necessary explanations to the article about its race-relatedness.
- Is it necessary to summarize the treatment strategies that can be implemented with the diagnosis of sarcopenia in a table?
- It is not appropriate to mention general statements without providing details. Therefore, it is necessary to summarize the effective ones in a table. For example, which supplement? Which diet, etc. Examples of this are frequently seen in the article "Some research [57,58] explores how prehabilitation—a combination of nutritional and exercise interventions before surgery—can improve muscle mass and thus enhance postoperative outcomes. This area shows promising potential for improving treatment results." Or "In the realm of prehabilitation and rehabilitation, studies have investigated the benefits of multimodal prehabilitation [111–113], nutritional status and the efficacy of nutritional supplements [114–116], as well as neuromuscular electrical stimulation (NMES) [117] in reducing postoperative sarcopenia. Besides complications and survival, researchers have also examined quality of life in post-operative CRC patients [118]. "
- For complex figures such as Figure 7, at least one paragraph should be included as a caption to explain the figure.
- It seems that the authors have focused more on the writing style and principles of bibliography than on the necessary information to support the title of the article. Therefore, it is necessary to present more on the results of the studies.
- I think it would have been better if the authors had conducted a systematic review, because they had a specific question that could be investigated with a systematic review, but if it had been accompanied by a meta-analysis, it would definitely have provided a more precise answer as an article.
Author Response

(The authors gave the same response as above.)

Reviewer 3 Report
Comments and Suggestions for Authors
- Is the literature review comprehensive? Include the cancer recent statistics at introduction part.
- Did you effectively balance quantitative data (such as statistical results) and qualitative insights (such as expert opinions or theoretical models)? Is the synthesis of these data clear and meaningful?
- While you mention significant research gaps, could you highlight any specific, under researched areas that would be impactful for advancing clinical practice? For example, are there emerging technologies, biomarkers, or imaging techniques that might help identify sarcopenia more accurately?
- Could the recommendations for future research be more specific in terms of methodologies or experimental designs? For instance, would large-scale, multicenter prospective studies be needed, or perhaps a focus on randomized clinical trials to address the efficacy of sarcopenia interventions?
- Author mention preoperative assessment of muscle condition, but could this be expanded by discussing specific methods of assessment? Are there validated tools (e.g., CT scans, bioelectrical impedance analysis, or other biomarkers) that should be prioritized in clinical practice?
- What would a personalized treatment model look like in the context of sarcopenia? Would it involve a tailored approach to nutrition, rehabilitation, or pharmacological interventions?
- Did you provide a thorough explanation of the methods used in VOSviewer and WordSift for readers unfamiliar with bibliometric analysis? For example, how were the thematic clusters identified, and were there any limitations to the analysis or its interpretation?
- While you mention the collaborative links among authors and countries, could you provide examples of successful collaborations or international initiatives that have driven advancements in this field?
- Could you expand on the MDT model and how it could be optimized in practice? Are there any barriers to effective collaboration between oncologists, surgeons, nutritionists, and physiotherapists?
- How can healthcare providers be better trained to recognize and address sarcopenia in CRC patients preoperatively? Do you suggest integrating sarcopenia assessment into standard CRC care guidelines?
- How confident are you in establishing causality between sarcopenia and poor postoperative outcomes in CRC? Do you address potential confounding factors that might skew the results of studies linking sarcopenia with surgical complications?
- The conclusion is insightful, but could it be further emphasized with a summary of the main recommendations for clinical practice and research? For example, a call for more interdisciplinary collaboration or a push for better standardization in sarcopenia diagnosis.
Author Response

(The authors gave the same response as above.)

Reviewer 4 Report
Comments and Suggestions for Authors
The manuscript presents a well-structured analysis of sarcopenia's impact on colorectal cancer surgical outcomes. Some sections, like the Introduction, could be made more concise by removing redundant explanations. Some figures could be better labeled or explained in captions. The manuscript uses visual summaries to present the research findings.
I expect to find some mention of the practical implications of the correct diagnosis of sarcopenia in clinical decision-making.
The temporal trend analysis is insightful, but it would be helpful to summarize why research on sarcopenia for patients with colorectal surged post-2020, possibly due to improved diagnostic imaging or increased awareness of frailty.
A section on potential intervention or treatment pathways could be of practical value.
As for language and grammar, a reduction of long sentences would improve readability.
Author Response

(The authors gave the same response as above.)

Round 2
Reviewer 2 Report
Comments and Suggestions for Authors
I have no further suggestion.